# Investigation of Energy Absorbed by Composite Panels with Honeycomb Aluminum Alloy Core

**DOI:** 10.3390/ma13245807

**Published:** 2020-12-19

**Authors:** Maciej Mogilski, Maciej Jabłoński, Martyna Deroszewska, Robert Saraczyn, Jan Tracz, Michał Kowalik, Witold Rządkowski

**Affiliations:** Institute of Aeronautics and Applied Mechanics, Warsaw University of Technology, 00-665 Warsaw, Poland; maciejmogilski@wutracing.pl (M.M.); maciejjablonski@wutracing.pl (M.J.); martyna.deroszewska@wutracing.pl (M.D.); robert.saraczyn@wutracing.pl (R.S.); jantracz@wutracing.pl (J.T.); michal.kowalik@pw.edu.pl (M.K.)

**Keywords:** composite panels, energy absorption, CFRP, perimeter shear, three-point bending

## Abstract

The aim of this study was to measure the energy absorbed by composite panels with carbon fiber-reinforced polymer (CFRP) skins and a 5052 aluminum alloy honeycomb core and to compare it to previous research and isotropic material—two 25 × 1.75 mm 1.0562 alloy steel tubes. The panel skins layup consisted of pre-impregnated Pyrofil TR30S 210 gsm 3K 2 × 2 twill oriented in directions 0/90 and −45/45 and having a consolidated thickness of 1 mm or 2 mm. The core consisted of a 15 mm or 20 mm honeycomb oriented along its lengthwise direction. The first test consisted of a three-point bending of specimens supported at a span of 400 mm with a 50 mm radius tubular load applicator in the middle. Second, a perimeter shear test was conducted using a 25 mm diameter punch and a 38 mm diameter hole. The results of the three-point bending test show that the energy absorbed by panels with 1 mm skins was similar to the energy absorbed by the tubes (96 J), which was better than the previously considered panels. In the case of perimeter shear, the average maximum forces for the top and bottom skin were 5.7 kN and 6.6 kN, respectively. For the panel with thicker skins (2 mm), the results were about 2 times higher.

## 1. Introduction

Composite panels are used in a variety of engineering sectors, such as aerospace, automotive, and civil [1,2,3,4]. This study continues the search for a panel that has sufficient energy absorption and mechanical properties to create a monocoque chassis that is compliant with the Society of Automotive Engineers’ (SAE) rules and regulations [5]. Based upon results of past work, which found that composite panels with carbon fiber-reinforced polymer (CFRP) skins and aluminum cores had insufficient characteristics to meet the requirements, two solutions were suggested.

The first suggestion was to change the type of material used to another type. Aluminum skins and cores were investigated by a few research teams with promising results [6]. There were also reports of other skins and cores that could be used, such as Glass Fiber Reinforced Polymer (GFRP), basalt, Nomex honeycomb, or foamcore [7,8,9,10]. The second option was to improve the existing panels. There have been numerous studies that have focused on such panels and ways to improve them [11,12,13,14,15,16]. Additionally, other types and specifications of aluminum honeycomb have shown promising results [17,18,19,20,21].

The composite panels investigated in this study were used to construct the chassis of a motorsport vehicle (monocoque). The main purpose of a chassis is to protect a driver from impacts and to serve as an attachment structure for other vehicle components. Obviously, all the loads that are created in these systems pass through the monocoque structure. This exhibits huge stresses and deformations on the chassis, which needs to have sufficient strength and stiffness to resist them. On the other hand, the structure must be lightweight, as it is a key requirement in motorsport racing. 

This is possible by using composite panels due to their high specific stiffness and specific strength [22]. The used material must meet the safety requirements imposed by the Society of Automotive Engineers (SAE) [5]. Therefore, our tests addressed the impact and bending of the composite panels and the energy absorbed in such a manner. Results and designs created by other research teams in this field can be found at [17,23,24,25,26,27,28,29,30].

The objectives of this study were:To determine how much energy the investigated composite panels can absorb in bending and compare the results with the ones obtained in previous research [31].To verify if the panels absorb at least the same amount of energy in bending as 25 × 1.75 mm 1.0562 alloy steel tubes, which can be used to construct the chassis (space frame) of a motorsport vehicle due to good welding and strength properties, as well as for economic reasons. This comparison is important from an application point of view since a monocoque chassis made from composite panels exhibits much higher torsional stiffness compared to a steel space frame, which is more favorable in motorsports.

## 2. Materials and Methods

### 2.1. Materials and Layup

Previous research [31] found the investigated panels to absorb less energy than steel tubes, which is unacceptable from a safety point of view. The skin thickness was about 1 mm with a 20 mm core, and this panel absorbed 59 J of energy while the two steel tubes obtained a result of 96 J. One solution to this problem would be to increase the thickness of the skins. However, thicker skins would increase the mass of the panels, thus making them unfavorable for the application, as the goal is to obtain a lighter or similar mass as the two steel tubes and at the same time to absorb more or an equal amount of energy. Having said that, it was also decided to test one panel that had thicker skins (2 mm) as a safety option to ensure that there was enough energy absorption.

Another solution was to change the layup and materials used. The layup has a significant impact on the stiffness and strength of a panel. Additionally, another requirement was to have a layup that was quasi-isotropic, meaning that the mechanical properties (stiffness, strength) in the main directions, i.e., −45, +45, 0, 90, must be the same. Obviously, in the case of woven material it can be assumed that orientations of −45/+45 and 0/90 are the same, as reinforcements run along both directions. Any number of plies can be used, yet thinner plies are much more expensive. Therefore, it was assumed that widely available materials should be used. A couple of layup types were considered, such as (−45, 45, 0, 90), (0, 90, −45, 45), (−45, 0, 90, +45) since the previous layup was (−45, 0, +45, 90). For this paper, it was decided to investigate the (0, 90, −45, 45) layup for the given panels as one of three options.

For the material it was decided that the skins of panels 1 and 2 would be made of four layers of 210 gsm out-of-autoclave (OOA) XPREG XC110 acquired from [32] oriented at (0, 90, −45, 45) from top to bottom. This is a pre-preg system based on Pyrofil TR30S 3K carbon fabric with a 2 × 2 twill weave with fibers oriented at 0 and 90 and pre-impregnated with an epoxy resin system in the b-stage. This was a Polyacrylonitrile (PAN) based carbon fiber with a filament diameter of 6.89 μm. Single ply weight was equal to 362 g/m^2^, with a fiber content of 58% (210 g/m^2^). The consolidated thickness of a single ply was equal to about 0.25 mm, and the final thickness of the laminate, consisting of 4 plies, was equal to about 1mm. This resulted in the layup, which had about 50% stiffness in the weakest direction and 60% strength compared to the main direction.

Additionally, one more panel was tested. Panel 3 utilized the same material, but a different layup. It consisted of 8 plies of the same material oriented at (−45, 45, 90, −45, 45, 90, 90, 90). This layup hopefully introduced a yield-like behavior as the top layers were oriented along a 45 direction. This was a safety option to fulfil the condition of energy absorption, even at the cost of a heavyweight panel.

For the core material, a 15 mm thick hexagonal aluminum alloy 5052 honeycomb, quired from [33], was used in the bending test. The honeycomb was oriented along the lengthwise direction, which was much stiffer than the widthwise direction and had higher shear strength. A perimeter shear test was performed using the same core material, yet the thickness was 20 mm. The orientation of the core in this test should not have been important. Parameters of core material are stored in Table 1.

To manufacture composite panels, it is important to follow standard practices together with the recommendations of the material manufacturer [32,34]. To cure the panels, a high-temperature curing cycle was used [35]. It is worth mentioning that panel 1 was produced using a two-step process, where the bottom skin was cured first, and then the honeycomb core and outside skin were laminated on top of it and cured during one cycle. To bond skins to the core material, an XPREG XA120 film adhesive, acquired from [35], was used: It is a structural adhesive with a ply thickness of 0.15 mm and a weight of 150 g/m^2^. To prepare the surface for bonding on the bottom skin, an 83 g/m^2^ Nylon 66 peel-ply, acquired from [36], was used during the curing process. Next, the surface was cleaned with water and oil-free compressed air and bonded with honeycomb. Since the top skin was bonded in one step, there was no preparation of the surface. Panel 2 was made in a one-step process where both the skins and the core were laminated together. It was done to check if there was a difference between those methods, and how large.

The test also compared the composite panels to a set of two 25 × 1.75 mm steel tubes made of 1.0562 alloy steel, which had a minimum yield strength of 335 MPa. The number of tubes was selected based on regulations [5] for conducting such tests and was based on the fact that the composite panel was replacing the structural wall of a chassis where such tubes were spaced ±275 mm apart. The panel had to absorb more energy over the distance of 19 mm than the steel tubes to be valid enough to apply in the monocoque chassis structure. Obviously, the failure modes of the two compared materials were different—panels fail in a brittle manner, whereas tubes made of mild steel are ductile and can deform plasticly, which creates a difference in energy absorption mechanisms as tubes can absorb significantly more energy in ductile deformation despite withstanding the significantly lower forces acting on them. The comparison was also useful to calculate and verify compliance of the testing machine and the test rig, which were made of the same steel type, yet with much thicker profiles. All specimens used in this test were placed in Table 2.

### 2.2. Test Machine Setup

All the tests were performed using an Instron 8516 testing machine acquired from [37]. The sampling frequency was equal to 1 kHz and the test velocity was set to 60 mm/min for both test types. The integral used to calculate energy absorbed was a one-dimensional numerical integral using a rectangular rule. It can be expressed as the force at a given step, multiplied by the change of displacement (i.e., the current position minus the previous one), and added to sum the energy from the previous step: (1)$$E_{i}={\mathrm{Force}}_{i}\times \left({\mathrm{Pos}}_{i}-{\mathrm{Pos}}_{i-1}\right)+E_{i-1}$$
where: $E_{i}$—Total energy absorbed at current time step (J), ${\mathrm{Force}}_{i}$—Force at given time step (kN), ${\mathrm{Pos}}_{i}$—position of a machine at a given time step (mm), ${\mathrm{Pos}}_{i-1}$—position of machine at previous time step (mm), $E_{i-1}\;$—Total absorbed energy from previous steps (J).

### 2.3. Three-Point Bending Test Setup

This test was performed using three tubular steel supports with a radius of 50 mm. The supports overhung the tested panel to prevent edge loading. The specimens, measuring 510 × 275 mm, were supported at a span distance of 400 mm from the bottom side of the rig. A single tube was placed above the specimen in the middle to introduce the bending load. Tubular supports were welded to the rig structure, but the connection to the lower part was realized with a bolted connection so that the horizontal alignment could be changed. The test setup is depicted in Figure 1. The supports were leveled with respect to ground level using a spirit level.

### 2.4. Perimeter Shear Test Setup

This test was performed using a 25 mm diameter flat punch aligned coaxially with a 32 mm diameter hole. Between these two objects, the tested composite panel was placed. The punch was mounted directly to the machine clamps, and the flat plate with a hole was laid on the lower part of the three-point bending test bed. The assembly is shown in Figure 2. Again, the test rig was aligned with respect to ground level.

### 2.5. Steel Tubes Test Setup

This test was performed using a three-point bending setup. Each tube was supported at a 400 mm span, and the load applicator was placed in the middle. The distance between tubes was 100 mm, and they were placed in equal distances from the symmetry axis to prevent uneven bending of the test rig. The setup is visible in Figure 3. Again, this test was done only to check if the panels absorbed a similar enough amount of energy to be considered equivalent to the steel tubes.

## 3. Results

The results were investigated in three categories: displacement, peak forces, and energy absorbed by the panel. The energy absorbed by the panel was calculated as a stepwise integral of the measured displacement difference and force value at a given step. The direct formula is available in the Methods section. Based on the test conducted with the steel tubes, the three-point bending test rig compliance was equal to 2994 N/mm. The compliance of the perimeter shear test bed was assumed to be two orders of magnitude higher, as there was no bending, just pure compression of the test rig members.

### 3.1. Three-Point Bending Test Results

The results are shown in Figure 4. A peak can be seen associated with the skin failure of the tested panels. The maximum loads acquired, as well as the energy absorbed by each panel and the steel tubes at displacement of 19mm are shown in Table 3. The curves of energy absorption can be seen in Figure 5.

The stress at failure was calculated as the force multiplied by the correctional area of a panel skin and divided by four times the moment of inertia. The obtained results were far from physical, as the stress levels were very low, and the failure happened due to face wrinkling. In most of the research (e.g., [12]) where such panels were investigated, the results have been presented as force/displacement or energy displacement rather than stress/strain; thus, a similar approach was also used in this paper.

### 3.2. Perimeter Shear Test Results

The obtained results are shown in Figure 6. Two force peaks can be seen—they are associated with a perforation of both composite panel skins: top and bottom. There was an additional peak in the linear elastic region at the beginning of the test. It is addressed in the Discussion section. Average maximum forces for panel 1 on the top and bottom skin were 5.69 kN and 6.56 kN, respectively, with a sample standard deviation of 0.24 kN and 0.546 kN, respectively. In the case of panel 3, only one sample was tested. The maximum forces acquired at each test sample are presented in Table 4.

Two additional lines are present in Figure 6—the front bulkhead support (FBHS) and side impact structure (SIS) requirements. They are important from an application point of view as they are the minimum requirement of the piercing force that a panel must withstand to be certified for usage in a chassis structure. The front bulkhead support (FBHS) is designed for the frontal part of a monocoque structure, which protects a driver’s legs. The side impact structure (SIS) protects the driver’s torso; thus, the requirement of peak force is higher.

## 4. Discussion and Conclusions

To sum up the experiments, it is important to compare the energy absorption in the three-point bending test (Table 3). As it can be seen, panel 2 had a similar absorption potential as the steel tubes, which, unlike panel 2, had insufficient absorption properties. However, the steel tubes achieved good energy absorption by yielding, which increased the energy absorbed after the tubes reached the yield limit. Such behavior was not found in the composite panels, where, after the skin failure, the force decreased to a value of around 2–3 kN. The failure occurred at the top skin as a result of local effects due to load application (face wrinkling). In panel 3, a slight plastic deformation was observed, probably due to the surface plies being oriented at 45 degrees which were carrying the load.

The difference between the energy absorbed by panels 1 and 2 could have resulted from the manufacturing technology. This implies that only panels made in a one-step process can be useful from an application point of view; however, the sample was quite small and would have required additional tests. Additionally, as can be noticed, the change of layup and material increased the energy absorption potential from 59 J in previous research to the current values of 74.7 J (panel 1) or 99.6 J (panel 2), even though the skin thickness (thus, the mass of a panel) had not increased, which is an important finding. Doubling the skin thickness increased the energy absorbed by approximately a factor of 2 (99.6 J to 202 J).

The energy absorption properties in three-point bending could be improved further, as suggested in the introduction, by adding another layer of interfacial reinforcement [16]. Another idea would be to test an alternative layup, where the top fibers are oriented at ±45 degrees instead of 0/90, which could possibly improve the energy absorption by introducing yield-like behavior at a cost of peak force, which was seen in panel 3. The last possibility would be to increase the thickness of the sandwich material, which would increase the cross-sectional moment of inertia. It would not increase the ultimate tensile strength of such a panel, but the stiffness could be higher.

The second experiment’s results show that the honeycomb structure began to crush at a force of about 2 kN (Figure 6). This corresponds with the honeycomb crush strength if one assumes that the area being crushed on the top of the panel is only under the 25 mm diameter punch. Both panels met the minimum requirement of 4 kN; however, it was only panel 3 that was sufficiently strong to reliably resist the 7.5 kN of shear force. The value of 2 kN at which the core began to yield can be important when designing any attachment points and for further investigations of inserts.

Doubling the skin thickness increased the peak forces by about a factor of 2, but not the absorbed energy as it depended greatly on the honeycomb between the skins. However, the average values fit the prediction, where each ply of 200 gsm fabric increased the force required to shear the skin by about 1.5 kN, which was roughly established from the results of previous studies. The perimeter shear test gave scattered results. The reason for the variation of results is unknown, but there might have been a non-uniform stress distribution forming under the composite panel that caused some of the panel skins to fail 10% sooner than the average. This non-uniform stress might have been due to test rig geometry or to the alignment of the specimens and the test rig. Additionally, from an application point of view, panels with four layers of fabric can be implemented, as the minimum requirement [5] was to withstand 4 kN at the highest peak for the front bulkhead support (FBHS). For the side impact structure, the panel 3 layup must be used. 

The next step will be to manufacture the monocoque chassis using panels that were successfully tested and verify the performance and safety of whole structure.

## Figures and Tables

**Figure 1 materials-13-05807-f001:**
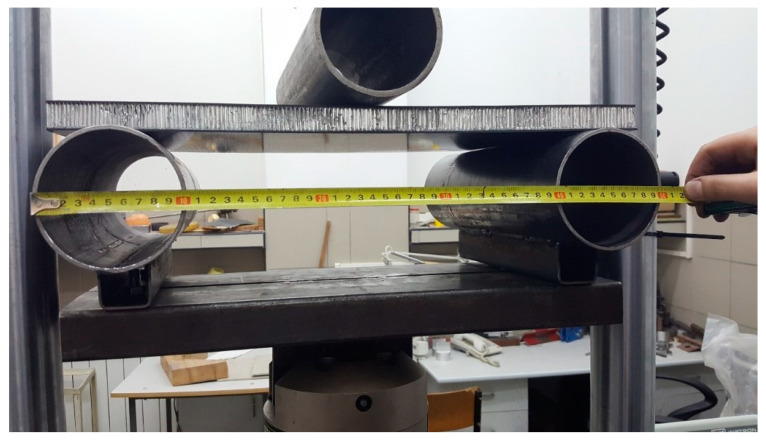
Three-point test setup.

**Figure 2 materials-13-05807-f002:**
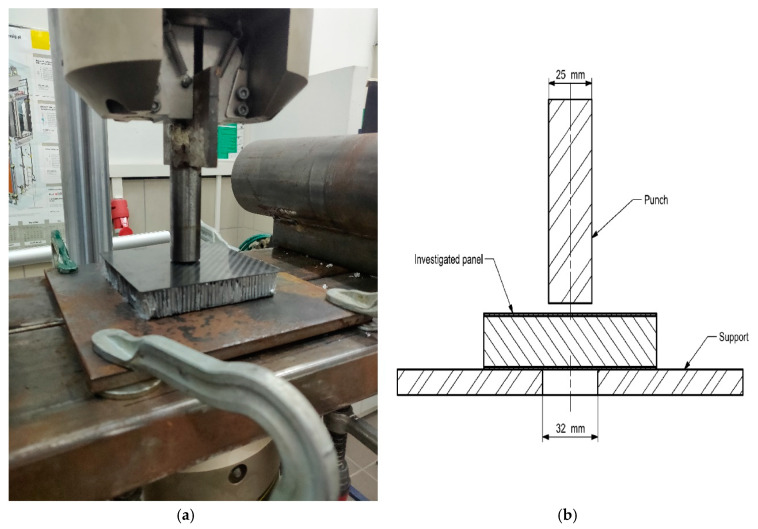
Perimeter shear test setup. (**a**) assembly on the test machine (**b**) cross-sectional view of the setup.

**Figure 3 materials-13-05807-f003:**
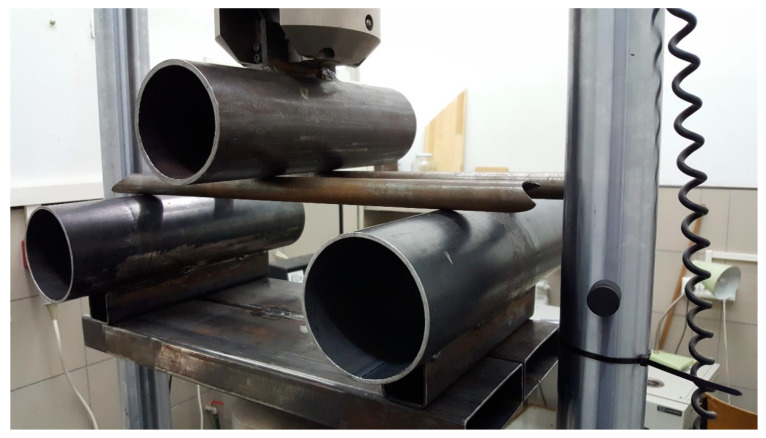
Steel tubes test setup.

**Figure 4 materials-13-05807-f004:**
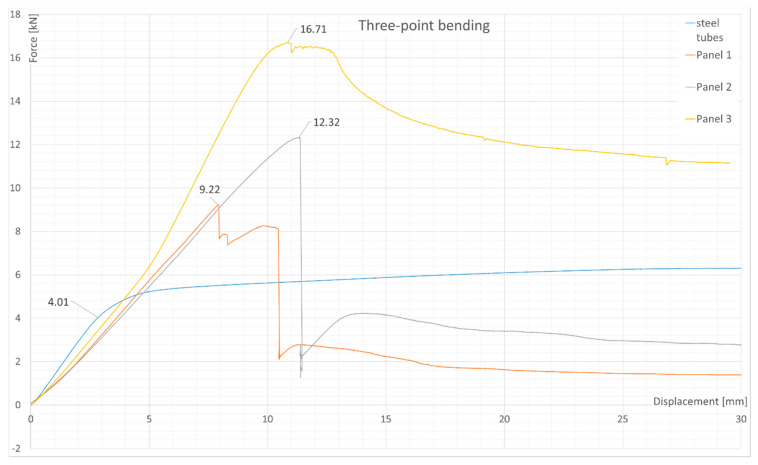
Three-point bending test results: Force as a function of displacement.

**Figure 5 materials-13-05807-f005:**
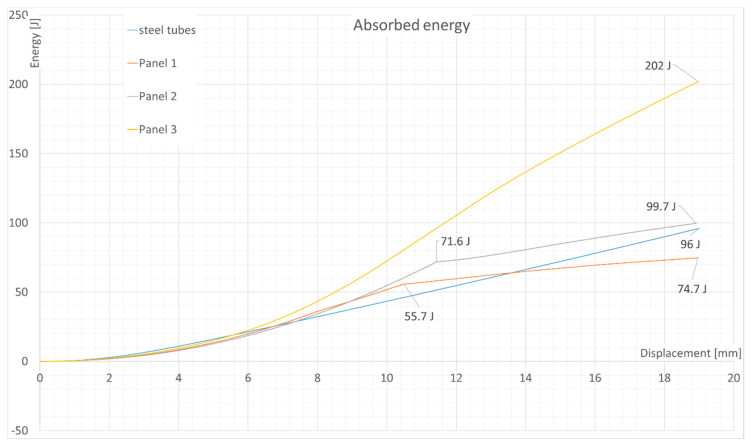
Three-point bending test results: Energy absorbed as a function of displacement.

**Figure 6 materials-13-05807-f006:**
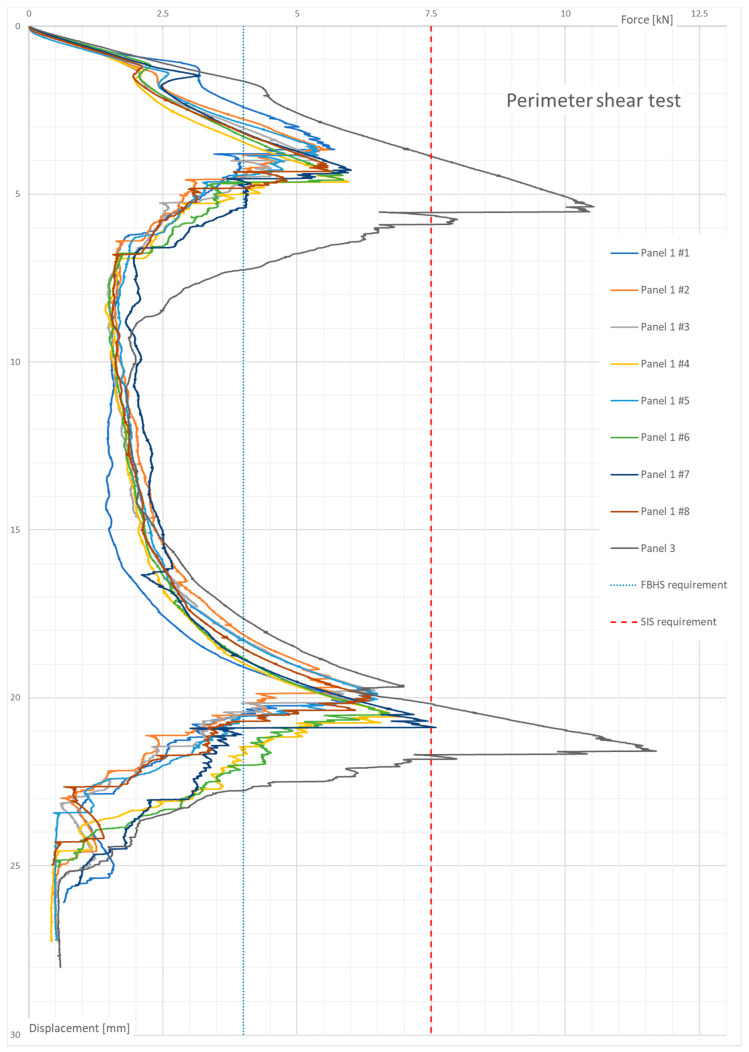
Perimeter shear test results.

**Table 1 materials-13-05807-t001:** Honeycomb core properties [33].

Property	Value
Cell size [mm]	3.2
Foil thickness/wall thickness [microns]	35
Plate Shear Modulus (lengthwise) [MPa]	482
Plate Shear Strength (lengthwise) [MPa]	2.34
Plate Shear Modulus (widthwise) [MPa]	214
Plate Shear Strength (widthwise) [MPa]	1.52
Compressive Strength (stabilized) [MPa]	3.85
Thickness [mm]	15 or 20

**Table 2 materials-13-05807-t002:** Tested panels and tubes properties.

	Panel 1	Panel 2	Panel 3	Steel Tubes (×2)
Number of 210 gsm plies	4	4	8	-
Layup	(0, 90, −45, 45)	(0, 90, −45, 45)	(−45, 45, 90, −45, 45, 90, 90, 90)	-
Core thickness (mm)	15	15	15	-
Weight (kg)	0.52	0.53	1.04	0.51
Technology (step process)	2	1	1	-

**Table 3 materials-13-05807-t003:** Results of three-point bending test.

	Panel 1	Panel 2	Panel 3	Steel Tubes
Maximum load (kN)	9.22	12.32	16.71	6.30
Energy absorbed at displacement of 19 mm (J)	74.74	99.66	202	96.01
Stress at failure (MPa)	222	297	148	391 *

* The stress given here is proof stress.

**Table 4 materials-13-05807-t004:** Results of perimeter shear test.

Test Sample	Panel 1	Panel 3
Peak Force—1st (kN)	$5.69_{-0.32}^{+0.31}$	10.52
Peak Force—2nd (kN)	$6.56_{-0.76}^{+1.04}$	11.69
Energy absorbed at displacement of 19 mm (J)	$64.66_{-5.66}^{+7.94}$	71.95

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
