# Peer review of "Investigation of Energy Absorbed by Composite Panels with Honeycomb Aluminum Alloy Core"

_materials, 2020, doi:10.3390/ma13245807_

Round 1

Reviewer 1 Report

  1. The subject is suitable for the journal and is relevant to current interests.
  2. The scientific objectives are inadequately stated. They should be stated in view of the prior related work and the open scientific questions to be addressed.
  3. Prior related work on CFRP and its sandwich composites is inadequately referenced.
  4. The results should be discussed in view of the prior related work.
  5. The force should be presented in terms of the stress. The displacement should be presented in terms of the strain.  The energy should be presented in terms of the energy per unit volume.
  6. The carbon fiber prepreg is inadequately described. The fiber diameter, the fiber volume fraction in the laminate, the number of fibers stacked along the thickness of a ply in the laminate, the thickness of a ply, and the carbon fiber type (PAN-based?) should be described. The commercial designation given does not provide satisfactory technical information.
  7. The film adhesive is inadequately described. The composition, thickness and method of application (for joining) should be described. The commercial designation given does not provide satisfactory technical information.
  8. The meaning of the following terms needs clarification: out-of-autoclave, w.r.t, stiffer direction, etc.
  9. The abstract is not adequate and is not self-contained. The description of the aluminum honeycomb core as “aluminum alloy sandwich” is confusing. The thicknesses of the skin and core should be mentioned. The meaning of some terms (XC110, stiffer direction) needs to be clarified.
  10. The composition and mechanical properties of the aluminum alloy should be reported.
  11. The title should be revised. The abbreviation CFRP should not be used. The honeycomb sandwich composite should be mentioned.
  12. The large difference in results between Panels 1 and 2 is odd, since these panels are supposedly identically prepared. These panels need to be described.  The large difference in results suggests inadequate reproducibility of the results.
  13. The meaning of the results obtained with the steel tubes alone is unclear in meaning. How can the steel tubes be tested without a panel? The steel tubes need more description.
  14. Data scatter information (± range based on the testing of multiple specimens of each type) is inadequate (Tables 1 and 2).
  15. The use of up to 5 significant figures in the values given in the Abstract seems excessive, due to the data scatter.
  16. Figure 1 is not useful and should be deleted in order to safe journal space.

Author Response

Dear Sir/Madam,

I am writing to inform that most of your suggestions have been addressed as they were very accurate. However, we have left few of them for further discussion:

3.Prior related work on CFRP and its sandwich composites is inadequately referenced.

This part will be added in the next iteration as unfortunately neither my colleague nor I were able to address the issue yet.

5.The force should be presented in terms of the stress. The displacement should be presented in terms of the strain.  The energy should be presented in terms of the energy per unit volume.

I totally agree that the results of mechanical properties should be represented in such way, however these tests do not address the tensile properties or directional properties which can be measured with strain gauges/digital correlation methods or simple strain equations. Obviously the strain can be measured as the change of initial position, yet it would rather be measurement of deflection.In reference [21] there was similar method of presenting the results.

Furthermore, there is a complex state of stress under the load applicator and it is difficult to determine the area at which the load is applied.We have calculated a stress within the skins in table 1, where sigma = F*cross-sectional area/(4*Moment of inertia) . However, as it can be seen the results are at least five times to small as carbon can withstand much more stress. Obviously, we can be wrong and any comments are appreciated.

The energy absorption per unit of volume is also difficult to measure since it is only a small part that is being deformed and relating this value to a whole panel could be misleading. Although it can be done. 

We hope that the rest of the issues was adequately addressed, if there are still some improvements to be made, please do not hesitate to make appriopriate comments.

Regards,
Authors

Reviewer 2 Report

Line 41-42 As state of the art regarding the optimization of the design of composite structures with particular attention to their energy absorption capability, the Authors may also consider the following references:

  • Wadley, H. N. (2006). Multifunctional periodic cellular metals. Philosophical Transactions of the Royal Society A: Mathematical, Physical and Engineering Sciences, 364(1838), 31-68.
  • Sarvestani, H. Y., Akbarzadeh, A. H., Niknam, H., & Hermenean, K. (2018). 3D printed architected polymeric sandwich panels: Energy absorption and structural performance. Composite Structures, 200, 886-909.
  • Casalotti, A., D’Annibale, F., & Rosi, G. (2020). Multi-scale design of an architected composite structure with optimized graded properties. Composite Structures, 252, 112608.

The manuscript repeats a testing protocol that has already been adopted for another very similar composite panel: the change of layup seems to be the only difference and it does not represent an interesting novelty. The authors should clearly state what are the proposed novelties to highlight their original contribution in this research field.

Line 48-49 It is not clear why the author decided to adopt that specific layup. Of course it is based on previous research, but a deeper and more comprehensive explanation about the reason why they focused on such layup sequence is required.

Line 51-52 the stiffness of the composite along various direction is mentioned and a figure is showed: it is not clear if the figure refers to the tested sample and how it has been determined. It this part of the results presented by the author or is it part of previous research? Please clarify.

Line 61 two steel tubes are tested to compare the structural performance, however it is not clear how the tubes are tested: the Author should explain which is the testing rig adopted for the tubes, in fact it is expected that it is considerably different from that adopted for the composite panel. Two additional pictures (companions of Fig. 2 and 3) of the testing rigs adopted for the tubes may clarify the testing procedure for the three-point-bending and shear test.

Figure 4 shows the force-displacement diagram of the composite and of the tubes, but due to the significant difference between the tested structures, the comparison should be conducted on different quantities such as maximum tensile stress in the components or strain. This will allow a more comprehensive reading of the results, while force-displacement is far too approximate and misleading, since it does not take into account the actual geometry of the tested samples.

Eq. (1) requires a more detailed explanation: the terms appearing in it are not defined, thus the utility of the formula is missed.

Line 103 the sudden drop in the force-displacement diagram of fig. 4 is said to be associated to skin failure: the author should clarify if it regards the top or bottom layer, allowing a deeper understanding on the type of failure (local effects due to load application, excess of compression in the top layer, excess of tensile stress in the bottom layer).

Author Response

Dear Sir/Madam,

I am writing to inform that most of your suggestions have been addressed as they were very accurate. However, we have left few of them for further discussion:

  1. Figure 4 shows the force-displacement diagram of the composite and of the tubes, but due to the significant difference between the tested structures, the comparison should be conducted on different quantities such as maximum tensile stress in the components or strain. This will allow a more comprehensive reading of the results, while force-displacement is far too approximate and misleading, since it does not take into account the actual geometry of the tested samples.

I totally agree that the results of mechanical properties should be represented in such way, however these tests do not address the tensile properties or directional properties which can be measured with strain gauges/digital correlation methods or simple strain equations. Obviously, the strain can be measured as the change of initial position, yet it would rather be measurement of deflection.

Furthermore, there is a complex state of stress under the load applicator and it is difficult to determine the area at which the load is applied. We have calculated a stress within the skins in table 1, where sigma = F*cross-sectional area/(4*Moment of inertia) . However, as it can be seen the results are at least five times to small as carbon can withstand much more stress. Obviously, we can be wrong and any comments are appreciated.

  1. Line 41-42 As state of the art regarding the optimization of the design of composite structures with particular attention to their energy absorption capability, the Authors may also consider the following references: […]

Thank you for recommendation, this part will be added in the next iteration as unfortunately neither my colleague nor I were able to address the issue yet.

We hope that the rest of the issues was adequately addressed, if there are still some improvements to be made, please do not hesitate to make appriopriate comments.

Regards,
Authors

Reviewer 3 Report

The study conducted three-point bending and shear tests on carbon fiber-skinned aluminum honeycomb panels, which has a good logical thinking. But it needs to be modified appropriately.

1)It is necessary to make a lot of supplements to the introduction, because many researchers have studied the structure of carbon fiber-skinned aluminum honeycomb panels, so a lot of previous work should be added to the introduction, and introduce the innovation of this research compared with the previous ones.

2)The experimental test schematic diagram should be re-photographed and arranged, and the test device should be highlighted. The front view angle is recommended.

3)All experimental results curves should be redrawn, the result curves should be coherent, and the information corresponding to the lines in the figure should be fully reflected. It is recommended to use ORIGIN software to draw.

4)The result discussion part should compare and analyze this research result and the existing research result, in order to reflect the superiority of this research result, reflect the innovation point.

Author Response

Dear Sir/Madam,

I am writing to inform that most of your suggestions have been addressed as they were very accurate. However, we have left few of them for further discussion:

2)The experimental test schematic diagram should be re-photographed and arranged, and the test device should be highlighted. The front view angle is recommended.

Unfortunately, due to this pandemic situation we had difficulties with that as there are restrictions within the building. Hopefully Figure 3 gives some perspective, we will probably add the CAD drawing of the setup. As far as apparatus is concerned it is a standard Instron 8516 machine.

We hope that the rest of the issues was adequately addressed, if there are still some improvements to be made to the figures, please do not hesitate to make appropriate comments.

The study conducted three-point bending and shear tests on carbon fiber-skinned aluminum honeycomb panels, which has a good logical thinking. But it needs to be modified appropriately.

1)It is necessary to make a lot of supplements to the introduction, because many researchers have studied the structure of carbon fiber-skinned aluminum honeycomb panels, so a lot of previous work should be added to the introduction, and introduce the innovation of this research compared with the previous ones.

We have updated the introduction and materials sections. The main innovation is new material used as well as layup type.

2)The experimental test schematic diagram should be re-photographed and arranged, and the test device should be highlighted. The front view angle is recommended.

We have added figure 3 which shows the setup of steel tubes 3 point bending test. We could additionally add photo of the whole machine, but for that we would have to perform some tests again and due to ongoing situation it was difficult.

3)All experimental results curves should be redrawn, the result curves should be coherent, and the information corresponding to the lines in the figure should be fully reflected. It is recommended to use ORIGIN software to draw.

We have corrected the figures, hopefully they are readable. If not, we will try to improve them using R software cause I’m unfamiliar with ORIGIN.

4)The result discussion part should compare and analyze this research result and the existing research result, in order to reflect the superiority of this research result, reflect the innovation point.

The problem with existing results is that

Regards,
Authors

Round 2

Reviewer 1 Report

1. Most of the points in the previous have not been adequately addressed.

2. The reply to the reviewers' comments should include the revised parts under each point of the review.  Please make it easier for the reviewer.

Author Response

To whom it might concern,

Please find the anwsers in the attached file.

Regards

Reviewer 2 Report

The revised version of the paper cannot be review until the authors provide a response to all of the comments. 

Author Response

To whom it might concern,

Please find the answers in the attachemnt

Regards
